# Cytokine Landscapes, Immune Dysregulation, and Treatment Perspectives in Philadelphia-Negative Myeloproliferative Neoplasms: A Narrative Review

**DOI:** 10.3390/jcm14176328

**Published:** 2025-09-08

**Authors:** Samuel B. Todor, Romeo Gabriel Mihaila

**Affiliations:** Faculty of Medicine, “Lucian Blaga” University of Sibiu, 550024 Sibiu, Romania; romeomihaila@yahoo.com

**Keywords:** myeloproliferative neoplasms, chronic inflammation, cytokine signaling, immune dysregulation, JAK2V617F mutation, checkpoint inhibitors

## Abstract

Philadelphia-negative myeloproliferative neoplasms (Ph-MPNs) are clonal hematologic malignancies characterized not only by driver mutations such as JAK2V617F, CALR, and MPL but also by a profoundly dysregulated immune microenvironment. Chronic inflammation and immune remodeling sustain malignant hematopoiesis and contribute to disease progression from essential thrombocythemia (ET) and polycythemia vera (PV) to overt myelofibrosis (MF). Pro-inflammatory cytokines and chemokines—including IL-2, IFN-α, IL-23, and TNF-α—drive abnormal T cell polarization, favoring a pathogenic Th17 phenotype. Lymphocyte subset analysis reveals a predominance of exhausted PD-1^+^ T cells, reflecting impaired immune surveillance. Concurrently, alterations in neutrophil apoptosis lead to persistent inflammation and stromal activation. GRO-α (CXCL1) is elevated in ET but reduced in MF, suggesting a subtype-specific role in disease biology. Fibrosis-promoting factors such as TGF-β and IL-13 mediate bone marrow remodeling and megakaryocyte expansion, while VEGF and other angiogenic factors enhance vascular niche alterations, particularly in PV. These immunopathologic features underscore novel therapeutic vulnerabilities. In addition to JAK inhibition, targeted strategies such as CXCR1/2 antagonists, anti-TGF-β agents, and immune checkpoint inhibitors (PD-1/PD-L1 blockade) may offer disease-modifying potential. Understanding the interplay between cytokine signaling and immune cell dysfunction is crucial for developing precision immunotherapies in MPNs.

## 1. Introduction

Hematopoietic stem cells self-renew and differentiate into myeloid or lymphoid lineages, forming mature blood cells such as RBCs, lymphocytes, granulocytes, megakaryocytes, and macrophages. This process is regulated by the bone marrow microenvironment, growth factors, and transcription factors [1].

Myeloproliferative neoplasms (MPNs) are a group of disorders characterized by abnormal proliferation of myeloid cell lines in the peripheral blood. First termed by William Dameshek in 1951, MPNs now include classic types—CML (chronic myeloid leukemia), PV (polycythemia vera), ET (essential thrombocythemia), and PMF (primary myelofibrosis)—CNL (chronic neutrophilic leukemia), CEL (chronic eosinophilic leukemia), and unclassifiable MPNs, per the WHO (World Health Organization) classification [2]. MPNs that lack the Philadelphia chromosome are classified as Philadelphia-negative MPNs, a distinct subgroup that excludes CML [3]. Prognosis varies among MPNs, but MF (myelofibrosis) generally shortens life expectancy more than PV or ET. A subset of patients may progress to a blast phase resembling acute myeloid leukemia, often resistant to standard treatment [4].

Despite their differences, MPNs share key clinical features, including similar bone marrow morphology, a predisposition to arterial and venous thrombosis, and a risk of progression to secondary MF or acute leukemia. These phenotypic similarities were recognized well before the discovery of activating mutations in Janus kinase 2 (JAK2), myeloproliferative leukemia (MPL), Calreticulin (CALR), and the identification of aberrant Janus kinase–Signal Transducer and Activator of Transcription (JAK-STAT) signaling, which has since refined the understanding of these disorders [5,6,7]. The JAK2 V617F mutation is identified in about 95% of patients with PV, while nearly all remaining cases carry mutations in JAK2 exon 12 [8]. In ET and PMF, approximately 50% of patients harbor the JAK2 V617F mutation, with most of the others exhibiting CALR or MPL mutations. A minority of ET and PMF patients are “triple-negative,” lacking all three mutations [9,10].

Following the acquisition of a key driver mutation—such as JAK2V617F, CALR, or MPL—a complex paracrine network involving inflammatory cytokines and growth factors significantly influences both the clinical presentation and clonal progression of MPNs. Notably, allogeneic stem cell transplantation (allo-HSCT) can eliminate the malignant clone and restore normal hematopoiesis, often resulting in resolution of chronic inflammation and regression of bone marrow fibrosis [11]. The connection between chronic inflammation and cancer is well established, playing a key role in both tumor initiation and clonal evolution [12,13].

Patients with PMF exhibit more pronounced immune dysfunction compared to those with ET or PV. The degree of immune impairment varies by driver mutation, with JAK2V617F and triple-negative patients showing greater dysfunction than those with CALR mutations. Additionally, patients with splenomegaly and overt PMF display more severe immune impairment, aligning with the known prognostic differences across MPN subtypes and clinical presentations [14].

This review aims to investigate the immune dysfunction and inflammatory dysregulation associated with the most common Philadelphia-negative myeloproliferative neoplasms (Ph-MPNs), including polycythemia vera (PV), essential thrombocythemia (ET), and primary myelofibrosis (PMF). We focus on analyzing key immune indicators such as lymphocyte subsets and cytokine profiles, correlating these findings with clinical data to better understand their impact on disease phenotype, progression, and prognosis. Additionally, this review explores emerging therapeutic strategies that specifically target the immune and inflammatory pathways implicated in Ph-MPNs. To achieve a comprehensive overview, a structured literature search was conducted across multiple databases, including PubMed/MEDLINE, Embase, Web of Science, and Scopus. Search terms combined keywords related to Ph-MPNs (“Philadelphia-negative myeloproliferative neoplasms,” “Ph-negative MPN,” “JAK2 mutation”), immune and inflammatory dysregulation (“immune dysfunction,” “inflammatory dysregulation,” “immune dysregulation”), the specific MPN subtypes (“polycythemia vera,” “essential thrombocythemia,” “primary myelofibrosis”), immune parameters (“lymphocyte subsets,” “cytokine profiling,” “immune markers”), and novel immunomodulatory therapies (“targeted therapies,” “immune modulators”).

The inclusion criteria encompassed original research articles and reviews published in English that reported immune parameters in PV, ET, or PMF patients, with relevant clinical correlations. Studies focusing on Philadelphia-positive MPNs (e.g., chronic myeloid leukemia), JAK2-negative MPNs, case reports, editorials, and abstracts without full data were excluded to maintain consistency. Given the relatively limited number of studies specifically addressing immune dysregulation in Ph-negative MPNs, we did not restrict the review to a defined timeframe or quantify the included papers but instead focused on relevance and scientific contribution.

## 2. JAK2-Mutation and Impact on Lymphocytes

The JAK/STAT signaling pathway was initially identified through studies on interferon (IFN)-induced transcription factor activation [15]. Key hematopoietic cytokines—such as interleukins (ILs), colony-stimulating factors (CSFs), IFN, erythropoietin (EPO), and thrombopoietin (TPO)—activate this pathway via type I homodimer receptors like EPO-R and TPO-R [16]. Cytokine binding induces receptor conformational changes that activate JAKs through phosphorylation. Phosphorylated JAKs provide binding sites for SH2-domain signaling molecules. Mutations in JAK2 cause constitutive activation of JAK/STAT signaling, particularly STAT3 and STAT5, contributing to hematopoietic malignancies alongside chromosomal abnormalities such as translocations and deletions [17,18,19].

STAT proteins reside in the cytosol and, upon phosphorylation by JAKs, dimerize and translocate to the nucleus to regulate genes controlling cell proliferation and survival, including cyclins and anti-apoptotic proteins [20]. Additionally, STAT dimers induce inhibitory proteins that terminate signaling by inactivating phosphorylated JAKs, their receptors, or preventing STAT-DNA binding [21].

Suppressors of Cytokine Signaling (SOCS) proteins are key negative regulators of the JAK/STAT pathway, consisting of an N-terminal domain, a central SH2 domain, and a C-terminal SOCS box. The family includes eight members: SOCS1–7 and CIS. Normally expressed at low levels, SOCS expression increases upon STAT activation [22].

They inhibit JAK/STAT signaling via two main mechanisms, by blocking JAK2 kinase activity through competition with STAT domains for receptor binding and by promoting proteasomal degradation of signaling proteins, thereby suppressing downstream pathway activation (Figure 1) [23].

Using a linear mixed effects model, it was observed that patients with JAK2V617F-mutated MPNs experienced a consistent decline in absolute lymphocyte count (ALC) over twice the normal age-related decrease seen in healthy elderly individuals [24]. The decline was even steeper in patients under 50, despite this age group typically maintaining stable ALC levels. Notably, this trend persisted regardless of cytoreductive therapy status suggesting that lymphopenia is a progressive and intrinsic feature of prolonged JAK2-V617F MPNs [25].

Transgenic mice with high JAK2V617F expression in HSCs showed marked lymphopenia due to a block in lymphoid development. This was linked to suppression of the Notch signaling pathway, essential for lymphocyte differentiation. Complementary findings, including Notch inhibition in human HSCs mimicking JAK2V617F effects, support the conclusion that JAK2V617F impairs lymphopoiesis by disrupting key developmental signals, particularly in advanced MPNs [26]. Notch signaling plays a key role in early lymphoid lineage decisions. Notch1 directs hematopoietic progenitor cells toward the T-cell lineage while suppressing B-cell development—its deletion leads to T-cell failure and ectopic B-cell formation in the thymus [27]. In contrast, Notch2 supports the differentiation of marginal zone (MZ) B cells, a specialized subset found in the spleen, distinct from follicular B cells in both function and surface markers [28].

In MPNs, impaired lymphocyte production becomes a key driver of immune dysfunction. As JAK2V617F-mutated HSCs outcompete normal ones, lymphopoiesis relies on a diminishing pool of healthy HSCs, leading to lymphopenia and weakened adaptive immunity. This strain may also contribute to the higher incidence of lymphomas, which notably arise from JAK2V617F-negative cells [29]. Moreover, the association between JAK2V617F mutation and increased risk of lymphoproliferative neoplasms (LPN) in MPN patients is notable. However, the detection of JAK2 wild-type lymphoid cells in some LPN cases suggests that JAK2V617F is not required for LPN development. Two explanations are plausible: either the MPN and LPN arise from separate progenitor cells, or both derive from a shared lymphoid-myeloid progenitor, with an initial genetic alteration followed by JAK2V617F mutation driving MPN and later mutations promoting LPN [30].

JAK2V617F driver mutation promotes a myeloid-biased hematopoiesis at the expense of lymphoid lineage commitment, leading to a marked reduction in lymphocyte production and a scarcity of JAK2V617F-positive lymphocytes. This effect is more severe in cases with bi-allelic mutations. The resulting lymphopenia contributes to an elevated neutrophil-to-lymphocyte ratio (NLR), a parameter increasingly recognized as a marker of immune dysregulation and poor prognosis in MPNs [31].

## 3. Lymphocyte Subsets Changes in MPNs

Regulatory B cells (Bregs), primarily defined by IL-10 production, modulate T-cell responses and suppress autoimmunity. They also exert IL-10–independent effects, notably by expressing PD-L1 (PDL1^+^-B-cells) to regulate follicular helper T cells (CD4^+^CXCR5^+^PD-1^+^) and dampen inflammation. Importantly, these cells resist anti-CD20 depletion and are key in maintaining humoral immune balance [32].

PD-1^+^ B cells, though rare in peripheral blood, are significantly increased in differentiated thyroid tumors and express high levels of PD-L1. Unlike typical IL-10-producing regulatory B cells, these PD-1^+^ B cells suppress CD4^+^ and CD8^+^ T-cell proliferation and viability primarily through PD-L1-mediated mechanisms. Their regulatory function is enhanced by continuous stimulation and reduced after tumor treatments like thyroidectomy and radioiodine therapy. This highlights their role in modulating T-cell responses within the tumor microenvironment [33].

In MF, T cells show increased activation and exhaustion, with elevated PD1 expression on both CD4+ and CD8+ subsets compared to healthy controls. T cells are skewed toward an effector phenotype, indicating a shift from quiescent to activated states. Treatment with the JAK inhibitor ruxolitinib reverses these aberrant T-cell distributions toward resting phenotypes. Baseline levels of CD4+ and CD8+ subsets correlate with monocyte and platelet counts, while their PD1+ fractions correlate with leukocyte counts and spleen size. Lower numbers of exhausted PD1+ CD4+ and CD8+ T cells are linked to better spleen size resolution and improved survival, suggesting that reduced T-cell exhaustion predicts favorable treatment outcomes [34].

PD-L1^+^ and PD-1^+^ B-cell proportions have been reported to be elevated in MF and PV patients compared to healthy controls, with some studies showing a decrease following treatment with ruxolitinib or combined therapy (ruxolitinib plus interferon alpha-2b) [35]. This suggests that the increased presence of these regulatory B-cell subsets may contribute to the immune suppression observed in MPNs. The reduction in PD-L1^+^ and PD-1^+^ B cells with therapy indicates a partial restoration of immune function, potentially improving anti-tumor immunity and reducing disease-related immune dysfunction (Figure 2) [36].

CD27 expression helps distinguish regulatory from effector T cells within the CD4+CD25+ population in inflamed tissues. Specifically, CD4+CD25+CD27+ cells are regulatory T cells (Tregs) characterized by high FoxP3 levels, lack of pro-inflammatory cytokine production, and strong suppression of T cell proliferation. In contrast, CD4+CD25+CD27– cells are effector T cells with low FoxP3, cytokine production, and no suppressive function [37]. In MPNs, regulatory T cells (Tregs) are typically decreased compared to healthy individuals, contributing to the chronic inflammatory state seen in these diseases. Treatment with JAK inhibitors often further lowers Treg levels but may simultaneously promote an increase in Th17 cells, suggesting a potential immune shift aimed at controlling the malignant clone [38].

Th17 cells play complex, paradoxical roles in tumor biology. They can promote tumor progression by driving angiogenesis, sustaining chronic inflammation, and enabling immune evasion via IL-17–mediated recruitment of neutrophils and myeloid-derived suppressor cells. Conversely, Th17 cells can also convert into IFN-γ-producing Th1-like cells, boosting cytotoxic T-cell activity and supporting tumor rejection, depending on the molecular context [39].

JAK inhibition with ruxolitinib has been reported to result in a decrease in circulating regulatory T cells (Tregs), which may partially impair immune surveillance and could contribute to the increased incidence of infections, including tuberculosis, herpes zoster, and viral pneumonia, observed in some treated MF patients [40]. Additionally, short-term treatment has been reported to reduce total CD3^+^ T cells, Tregs, Th1, and Th17 populations, accompanied by decreased production of pro-inflammatory cytokines, including TNF-α, IL-5, IL-6, and IL-1β. These findings suggest a potential broad immunomodulatory effect on CD4^+^ T cell function [38]. Reports suggest that JAK inhibitors initially suppress CD4^+^ T-cell function and cytokine production, followed by a shift in the immune profile that may favor a Th17-dominant response. This dual effect could potentially contribute to the increased risk of atypical infections observed in some treated patients [41].

In MF, distinct mutations (JAK2V617F, CALR, and triple-negative) have been associated with specific alterations in immune cell subsets. Across all patient groups, reductions in dendritic cells and impaired monocyte-to-DC differentiation have been observed. In JAK2-mutated patients, studies report lower levels of Th17 cells, myeloid DCs, and effector Tregs, alongside higher frequencies of ILC1 and cytokine-producing Tregs. CALR-mutated patients appear to have elevated ILC3, reduced Th1 cells, and Tregs with functional impairments linked to increased proliferation. In triple-negative patients, decreased CD3^+^ T cells, effector Tregs, and Th1 cells have been described, with a relative increase in ILC1. These observations suggest mutation-specific patterns in immune dysregulation, although further studies are needed to confirm these trends across larger cohorts [42].

Evidence suggests that the JAK2 mutation may be associated with shifts in lymphocyte subsets, as reflected by the ability of JAK2 inhibitors to modulate the immune landscape. These agents have been reported to affect lymphocytes, including Th1, Th17, and Tregs, as well as dendritic cells (DCs) and myeloid-derived suppressor cells (MDSCs). Importantly, available data did not show statistically significant differences among ruxolitinib, momelotinib, and pacritinib in terms of their effects on T helper cell differentiation or T cell proliferation in CFSE assays; however, whether these findings reflect a true class-wide effect of JAK2 inhibition remains to be further validated [43].

Natural killer (NK) cells are innate lymphocytes with key roles in tumor and viral defense through cytotoxicity and cytokine secretion, such as IFN-γ and TNF-α, which influence immune cell development and function [44]. They are crucial in controlling metastatic spread, with low NK cell numbers correlating with advanced disease and impaired prognosis, while strong NK cytotoxicity is linked to better outcomes [45]. NK cells are classified into two main subsets: CD56 ^dim^ CD16^+^ (cytotoxic, ~90% of circulating NKs) and CD56 ^bright^ CD16^−^ (regulatory, cytokine-producing) [46,47,48]. Untreated patients with MPNs had the lowest NK cell counts [49].

Ruxolitinib-mediated suppression of the JAK/STAT pathway has been reported to affect natural killer (NK) cells, with studies describing a reduction in NK cell numbers, predominantly due to impaired maturation, as reflected by an increased ratio of immature to mature NK cells [50]. Functionally, NK cells exhibit decreased cytotoxic activity, including reduced IFN-γ production and defective lytic synapse formation with target cells [51]. Mechanistically, these effects appear to be largely indirect, potentially resulting from ruxolitinib-induced modulation of dendritic cell function and decreased secretion of cytokines such as IL-12 and IL-15, which are important for NK cell differentiation and activation. Notably, these alterations have been reported to be reversible upon treatment discontinuation, with NK cell maturation and cytotoxic function recovering. Clinically, these findings may contribute to the increased susceptibility to infections observed in some patients receiving ruxolitinib [52]. Table 1 summarizes the lymphocyte subsets in MPNs and treatment impact.

## 4. Impact of JAK2-Mutation on Granulocytes

Neutrophils expressing JAK2-V617F, but not CALR mutations, drive inflammation in MPN by producing pro-inflammatory cytokines and evading clearance through upregulation of the “don’t-eat-me” signal CD24 via GM-CSF-JAK2-STAT5 signaling. These CD24^hi neutrophils infiltrate megakaryocytes through emperipolesis, enhancing active TGF-β production, which in turn promotes platelet overproduction and MF (myelofibrosis). Targeting CD24 restores neutrophil clearance, reduces pathogenic neutrophil–megakaryocyte interactions, and mitigates disease progression in preclinical models, positioning CD24 as a promising innate immune checkpoint and potential therapeutic target in MPN. Further studies are needed to evaluate its safety and efficacy in humans [53,54].

Neutrophils in MPN play a key pro-inflammatory and pathogenic role. JAK2V617F-mutant neutrophils promote inflammation via cytokine production (e.g., IL-1β, IL-8), resist apoptosis, and contribute to disease progression. NF-E2 (nuclear factor, erythroid 2), elevated in MPN, is induced in neutrophils and promotes IL-8 secretion, leading to neutrophilia and stem cell mobilization. Neutrophils are also a major source of IL-8 in MF. Biomarkers such as YKL-40 and NGAL (lipocalin-2), which correlate with neutrophil counts and JAK2V617F burden, are elevated, with NGAL enhancing malignant HSC survival and fibrosis [55].

Neutrophil extracellular traps (NETs) are web-like structures composed of nucleic acids, histones, enzymes, and antimicrobial proteins, which are expelled to immobilize and destroy pathogens [56]. While they play an essential role in host defense and innate immunity, NETs are also implicated in sterile inflammation. The process of NETosis involves neutrophils releasing decondensed DNA together with nuclear proteins and enzymes such as myeloperoxidase into the extracellular milieu. Beyond their antimicrobial function, NETs influence platelet biology; experimental data have shown that NETs, particularly when stimulated by lipopolysaccharides, can trigger platelet aggregation and clustering [57]. This underscores the contribution of neutrophil-driven mechanisms to thrombus formation. Although the body of evidence on NETosis in MPNs is still modest, animal studies suggest that increased JAK2 kinase activity enhances NET release, whereas inhibition of this pathway through targeted therapies mitigates thrombotic risk [58].

Bone marrow fibrosis in PMF is a multifactorial process involving both hematopoietic and stromal cell dysregulation. Emerging evidence highlights the role of monocytes harboring the JAK2 V617F mutation in fibrogenesis, acting beyond their classical immune functions [59].

Fibrocytes were identified in the bone marrow of PMF patients [60]. These cells contribute to extracellular matrix (ECM) deposition and are implicated in the fibrotic process. Transplantation of PMF bone marrow into immunodeficient mice induced a lethal fibrotic phenotype, which was attenuated by treatment with pentraxin 2, an inhibitor of fibrocyte differentiation [60]. This suggests a critical role of monocyte-derived fibrocytes in disease progression.

To better understand these mechanisms, a monocytic cell line stably expressing JAK2 V617F and a counterpart with JAK2 wild-type (wt) overexpression were established [59]. These models revealed significant alterations in the expression of key pro- and anti-fibrotic factors. Specifically, MMP-13 expression was upregulated, while MMP-9, MMP-12 and TIMP-1 levels were reduced, indicating an imbalance in MMP/TIMP homeostasis typical of fibrotic tissue [61,62,63]. Although TIMP-2, TIMP-3, and TIMP-4 expression remained unchanged, the altered enzyme/inhibitor profile suggests disrupted ECM turnover.

## 5. Megakaryocytes at the Core of Myelofibrosis

Disordered megakaryocyte (MK) biology stands at the heart of MPN pathogenesis and plays a pivotal role in histopathologic differentiation, especially between PMF, including its early (pre-fibrotic) stage, and ET. In diagnostic practice, pathologists rely heavily on the nuances of MK architecture—such as size, clustering patterns, and nuclear features—to distinguish PMF from ET [64]. Bone marrow (BM) examination is essential for accurately diagnosing ET and distinguishing it from other myeloid neoplasms, particularly pre-PMF. In ET, BM is often normocellular with increased, large to giant, hyperlobulated megakaryocytes arranged in loose clusters. In pre-PMF, megakaryocytes show abnormal maturation with hyperchromatic, irregularly folded nuclei, tight clustering, increased nuclear-to-cytoplasmic ratio, and bulbous nuclear appearance, often accompanied by granulocytic proliferation and mild reticulin fibrosis [65]. The presence of anemia, leukocytosis, splenomegaly, and elevated LDH further favors a pre-PMF diagnosis, and considering these features alongside marrow morphology enhances diagnostic accuracy and prognostic assessment [66].

Among the classic MPNs, MF most prominently exhibits megakaryocytic dysregulation, typified by an overproduction of MKs that fail to mature appropriately. This developmental derailment manifests clinically through fluctuations in platelet counts, morphological anomalies such as the presence of abnormally large platelets, and functional impairments in aggregation responses—features that may mimic congenital platelet function disorders [67,68,69]. Marrow fibrosis can be a downstream consequence of such megakaryocyte dysfunction, positioning MF within a broader spectrum that also encompasses inherited megakaryocytic and platelet pathologies [70].

Experimental models have shed light on the contributory role of mutant MKs in MPN initiation and evolution. Using mice engineered to express the JAK2V617F mutation specifically in the MK lineage via Pf4-Cre recombinase, it was proven that MKs not only disrupt the homeostasis of the surrounding hematopoietic environment but also drive erythro-megakaryocytic proliferation and a PV-like phenotype. Importantly, these MKs are also potent sources of inflammatory and chemotactic mediators—including IL-6, CCL11, CXCL1, and CXCL2 [71].

In MF, MKs are considered the primary source of fibrogenic signals that orchestrate the activation of stromal fibroblasts, extracellular matrix remodeling, neovascularization, and abnormal bone remodeling. Key pro-fibrotic molecules released by MKs include transforming growth factor-beta (TGF-β), platelet-derived growth factor (PDGF), fibroblast growth factor (FGF), vascular endothelial growth factor (VEGF), thrombospondin, CXCL4, various macrophage inflammatory proteins. Among these, the roles of TGF-β and PDGF/PDGFR axis have been most extensively investigated in mediating marrow fibrosis [72].

## 6. Distinct Cytokine Profile and Impact on MPNs Progression

PMF is immunobiologically distinct from ET and PV, exhibiting a uniquely amplified cytokine signature that reflects its fibrotic and inflammatory pathogenesis. Elevated levels of interleukin-2 (IL-2), IL-2 receptor (IL-2R), and IL-6 denote an activated T-cell and inflammatory microenvironment [73], while increased levels of IL-12, IL-17, TNF-α, and interferon-α (IFN-α) suggest enhanced differentiation of Th1 and Th17 cells, contributing to systemic immune dysregulation [74,75].

On the anti-inflammatory axis, IL-1RA, IL-4, and IL-10 are paradoxically upregulated, suggesting a compensatory immunoregulatory response to chronic inflammation. Notably, IL-10 has been shown to suppress myelopoiesis in MF by significantly inhibiting autonomous CFU-GM formation in vitro, indicating a potential regulatory role in controlling pathological myeloid proliferation [73,74,76]. PMF also displays significantly increased levels of chemokines such as MIP-1β, RANTES, and variably elevated MCP-1, which may perpetuate myelomonocytic recruitment and stromal activation [73,74,77]. Notably, inherited host genetic factors such as the MCP-1 -2518 A/G SNP (rs1024611) have been associated with secondary MF and correlate with more severe hematologic profiles, including higher IPSS risk categories and dysregulated chemokine production, suggesting a functional link between germline variation, inflammatory signaling, and disease progression. This could be due to the fact that MCP-1 is a key mediator of monocyte recruitment, driving inflammatory responses, oxidative stress, and fibro-angiogenesis, all of which are pathophysiological hallmarks of MF [78].

Elevated FGF, thrombopoietin (TPO), and TGF-β highlight the pro-fibrotic signaling cascade that drives marrow remodeling and disease progression [73,79]. Recent evidence also implicates IL-13 as a key driver of disease evolution in MF. IL-13 not only promotes the expansion of mutant megakaryocytes but also enhances TGF-β surface expression and collagen biosynthesis, reinforcing fibrotic transformation. Elevated IL-13 levels and increased IL-13 receptor expression in bone marrow megakaryocytes of MF patients and murine models suggest that the IL-13/IL-4 axis may represent a novel therapeutic target for mitigating fibrosis and slowing disease progression [80].

Collectively, this cytokine profile—marked by both immune activation and stromal reprogramming—supports the concept of PMF as a cytokine-driven malignancy, distinct from other BCR-ABL-negative MPNs.

In PV, a subset of inflammatory and angiogenic mediators exhibits selective upregulation, distinguishing this MPN subtype from both ET and PMF. Specifically, IL-7 and IL-23 were found to be elevated exclusively in PV. Notably, plasma levels of IL-23 were significantly increased in patients with PV compared to healthy controls, while no significant difference was found between ET and controls, supporting a PV-specific upregulation pattern [73]. IL-23 acts primarily on cells of the innate immune system and enables the maintenance and expansion of Th17 cells. While it does not induce Th17 differentiation or IL-17A production from naïve CD4^+^ T cells on its own—due to the absence of substantial IL-23 receptor (IL-23R) expression—IL-23 plays a crucial role in sustaining the pro-inflammatory Th17 axis, contributing to ongoing immune activation in PV [81,82], this finding supports the important role of Th17 cells in the pathology of MPNs.

Among growth factors, vascular endothelial growth factor (VEGF) showed increased levels in PV and PMF, but remained unchanged in ET, indicating PV-specific pro-angiogenic activity [73,83,84]. Mechanistically, VEGF promotes a positive feedback loop involving chronic inflammation and angiogenesis, particularly in PV. In patients with PV, VEGF upregulated the expression of key angiogenic mediators, including endothelial nitric oxide synthase (eNOS) and hypoxia-inducible factor-1α (HIF-1α), an effect potentiated by IL-6 and amplified under JAK1/2 inhibition (e.g., Ruxolitinib). Interestingly, while VEGF enhanced the gene and protein expression of eNOS and HIF-1α in PV mononuclear cells (MNC), it had the opposite effect in PMF, where it suppressed their expression. This differential regulation underscores disease-specific angiogenic signaling, with PV exhibiting a VEGF-driven, inflammation-mediated angiogenic axis, potentially contributing to its thrombo-inflammatory phenotype. Moreover, VEGF stimulation increased IL-6–positive MNCs, suggesting a VEGF–IL-6 feed-forward loop that reinforces vascular remodeling in PV [85].

In PV, hepatocyte growth factor (HGF) is overexpressed by bone marrow stromal cells and erythroblasts, promoting an autocrine/paracrine loop that supports aberrant erythropoiesis. HGF induces IL-11 production, and both cytokines converge on STAT3 activation, contributing to the proliferation of JAK2V617F-mutated cells. Notably, the upregulation of HGF and IL-11 appears independent of the JAK2V617F mutation itself, suggesting a parallel, mutation-independent pathogenic pathway in PV [86,87].

GRO-α (CXCL1), a chemokine implicated in neutrophil recruitment and inflammatory amplification, emerges as a relevant biomarker in the pathophysiology of MPNs, particularly in ET. GRO-α was found to be selectively elevated in ET, while remaining unchanged in PV and PMF. This restricted pattern of elevation suggests a unique role for GRO-α in the inflammatory microenvironment characteristic of ET [88].

The source of GRO-α appears to differ across MPN subtypes. In ET, GRO-α is predominantly released by CD56^+^CD14^+^ pro-inflammatory monocytes, reflecting a monocyte-driven inflammatory state. In contrast, transcriptomic data indicate that platelets from MF patients exhibit a significant upregulation of CXCL1 mRNA, implying that platelets become a major GRO-α source in later disease stages. This shift may reflect the evolving nature of cytokine signaling in MPNs, moving from immune-cell derived to megakaryocyte/platelet-driven pathways during fibrotic progression [89].

Despite the elevated CXCL1 transcripts in MF platelets, circulating GRO-α levels in MF are paradoxically lower than in ET. This may be explained by defective platelet granule release, reduced activation capacity, or platelet exhaustion, all of which are common in advanced myeloproliferative disease. Such findings support the hypothesis of a dysfunctional platelet-monocyte axis contributing to chronic inflammation and disease evolution in MPNs [89].

In ET, bone marrow mesenchymal stromal cells (BM-MSCs) exhibit significantly reduced secretion of stem cell factor (SCF), a key hematopoietic growth factor. This deficiency contributes to impaired support of hematopoiesis, evidenced by the reduced expansion of CD34^+^ hematopoietic stem/progenitor cells and diminished long-term hematopoietic capacity. Additionally, low SCF correlates with a disrupted bone marrow microenvironment, including reduced secretion of IL-6 and VEGF, and contributes to immune dysregulation. Specifically, BM-MSCs from ET patients show impaired suppression of PBMC proliferation and dysfunctional promotion of regulatory T cell (Treg) differentiation. Thus, low SCF plays a central role in the hematopoietic and immunoregulatory deficiencies seen in ET [90].

Table 2 summarizes all the cytokine patterns associated with MPNs.

## 7. JAK Inhibitors in Ph-Negative MPNs

Ruxolitinib is an oral JAK inhibitor that selectively targets JAK1 and JAK2. It was initially approved by the FDA in 2011 for the treatment of intermediate- and high-risk PMF, based on the pivotal COMFORT-I and COMFORT-II trials, representing the first therapy specifically indicated for this population. Later, in 2015, its approval was extended to patients with PV who are intolerant or resistant to hydroxyurea. Additionally, ruxolitinib is approved for steroid-refractory acute graft-versus-host disease [91,92]. Ruxolitinib outperforms best available therapy in PV, improving hematocrit control, treatment response, and symptoms while reducing thromboembolism, but increases anemia, herpes zoster, and skin cancer risk [93]. Furthermore, it provides rapid and durable improvement of splenomegaly and symptoms in PMF, offering a survival benefit regardless of mutation status, though careful dose titration and monitoring are needed due to cytopenias and infection risk [94].

Fedratinib, a JAK2-selective inhibitor approved in 2019 for intermediate- to high-risk primary or secondary myelofibrosis, reduces spleen volume and improves symptoms in 30–40% of patients. It is particularly useful for those resistant or intolerant to ruxolitinib. Common adverse effects include anemia, gastrointestinal symptoms, and elevated liver or pancreatic enzymes; rare Wernicke encephalopathy requires thiamine prophylaxis [95].

Pacritinib, approved in 2022 for intermediate- to high-risk myelofibrosis with severe thrombocytopenia (<50 × 10^9^/L), effectively reduces spleen volume and alleviates symptoms even in patients with low platelet counts [96]. Clinical trials, including PERSIST-2, and real-world data support its efficacy and full-dose administration regardless of thrombocytopenia [97]. Common adverse effects include diarrhea, nausea, and vomiting, with additional risks of bleeding, QT prolongation, thrombocytopenia, and elevated liver enzymes [98].

Momelotinib, approved in September 2023 for anemic patients with high- or intermediate-risk myelofibrosis (PMF or secondary MF post-PV/ET), improves anemia-related outcomes and overall quality of life [99,100]. Meta-analyses indicate good efficacy without increased adverse events, though heterogeneity in control treatments limits direct comparisons [101].

## 8. Future Perspectives in MPNs Treatment

Checkpoint inhibition with pembrolizumab was explored in a phase 2 study enrolling patients with intermediate-2 or high-risk primary or secondary MF who were ineligible for or refractory to ruxolitinib. Although pembrolizumab was well tolerated, no objective clinical responses were observed, leading to early study termination. However, immune profiling revealed signs of immune activation, including increased T-cell receptor diversity and activation markers, suggesting partial reversal of T-cell exhaustion. These findings indicate that PD-1 blockade alone is insufficient to produce clinical benefit in MF but may enhance immune function [102].

Despite promising preclinical data supporting PD-1/PD-L1 blockade in myelofibrosis, clinical experience has been disappointing. In a Phase II study of nivolumab in eight MF patients, no objective responses were observed; five achieved only transient disease stabilization (median 3.3 months), with a median overall survival of 6.1 months, leading to early study termination. Treatment was generally well tolerated, but efficacy endpoints were not met [103].

Vaccination with PD-L1–derived showed an enhanced anti-regulatory immunity response in MPN, as spontaneous CD4^+^ T cell responses against PD-L1 were detected in 71% of patients. These responses were significantly diminished in advanced versus non-advanced MPN. However, these results stem from preclinical, immunological exploratory data and require clinical confirmation and formal safety assessment before therapeutic implementation [104].

Reparixin, a CXCR1/2 inhibitor, demonstrated a reduction in bone marrow and splenic fibrosis in GATA1^low^ mouse models, without apparent effects on peripheral blood counts. The degree of fibrosis reduction correlated inversely with plasma reparixin levels. These findings suggest that inhibition of CXCL1–CXCR1/2 signaling may modulate TGF-β expression in megakaryocytes and contribute to antifibrotic effects, supporting the potential of CXCL1 blockade as a therapeutic strategy in myelofibrosis, pending clinical evaluation [105]. Selective CXCR2 antagonism with SB272844 effectively reversed the anti-apoptotic effects of GRO-α and, to a lesser extent, IL-8 on human neutrophils. This indicates that GRO-α acts primarily through CXCR2, while IL-8 signals through both CXCR1 and CXCR2. CXCR2 blockade may therefore restore neutrophil apoptosis and help resolve neutrophil-driven inflammation, supporting its potential therapeutic use in inflammatory conditions, including MPNs [106]. Dual CXCR1/2 antagonists (e.g., SCH527123) inhibit neutrophil functions (chemotaxis, degranulation, ROS production) more effectively than selective CXCR2 blockers (e.g., SB265610), particularly in response to CXCL8. While dual blockade shows greater in vitro anti-inflammatory efficacy, in vivo confirmation is limited due to species-specific CXCR1 differences. Dual CXCR1/2 inhibition may offer superior therapeutic benefit in neutrophil-driven lung diseases such as COPD, asthma, and bronchiectasis [107].

IL-23 acts as a pro-carcinogenic cytokine by promoting inflammation, angiogenesis, and tumor progression through JAK/STAT pathway activation and IL-23R upregulation. Its activity supports Th17 cell expansion, which may counteract the anti-tumor effects of IFN-γ–producing Th1 cells, ultimately facilitating immune evasion and malignancy [108,109]. In psoriasis, selective IL-23p19 inhibitors (guselkumab, tildrakizumab, risankizumab) have demonstrated high efficacy by disrupting this axis. IL-23 promotes expansion of pathogenic T cells producing IL-17A, IL-17F, IL-6, and TNF, contributing to chronic inflammation [110]. Notably, PV exhibits elevated IL-23 levels, implicating this cytokine in MPN-associated inflammation [111]. Preclinical data show that anti–IL-23p19 therapy suppresses inflammatory cytokines and prevents disease relapse in autoimmune models. These findings support IL-23 blockade as a promising therapeutic strategy in PV and other MPNs [108].

Aurora kinase A inhibition has emerged as a promising therapeutic strategy in MF. Alisertib, a selective Aurora A inhibitor, has demonstrated the ability to induce apoptosis in abnormal megakaryocytes, leading to antifibrotic and antitumor effects in MPN cells. Treatment with alisertib resulted in normalized megakaryocyte morphology and a reduction in bone marrow fibrosis [112].

TGFβ inhibition with AVID200, a TGFβ1/3 protein trap, was well tolerated in a phase 1b trial of advanced MF patients resistant or intolerant to ruxolitinib. While clinical responses based on IWG/MRT criteria were limited, AVID200 significantly reduced circulating TGFβ1 levels and improved platelet counts in most patients. Notably, two patients normalized platelet levels, and 17 showed increases from baseline. No major changes in bone marrow fibrosis or megakaryocyte morphology were observed [113,114].

In the ACE-536-MF-001 trial with 95 MF patients, luspatercept improved anemia and reduced transfusion burden across both transfusion-dependent and non-dependent cohorts. Hemoglobin increases ≥ 1.5 g/dL were seen in up to 50% of non-transfusion-dependent patients, while about half of transfusion-dependent patients had ≥50% fewer transfusions. Symptom scores also improved, especially in patients on stable ruxolitinib [115]. Sotatercept, an activin receptor IIA ligand trap targeting TGF-β superfamily members that inhibit erythropoiesis, was evaluated in a phase II trial for anemia in MF patients, both transfusion-dependent (TD) and non-transfusion-dependent (NTD). Administered subcutaneously every 3 weeks, either as monotherapy or combined with stable ruxolitinib, sotatercept showed anemia responses defined by sustained hemoglobin increases (≥1.5 g/dL) or transfusion independence [116].

Tagraxofusp (SL-401) is a fusion protein combining IL-3 with a truncated diphtheria toxin, specifically designed to target CD123. Upon binding to CD123, the agent is internalized via endocytosis, ultimately inducing cell death [117]. In a clinical study involving MF patients, tagraxofusp led to symptom improvement and spleen reduction in nearly half of participants. A phase I/II trial (NCT02268253) is underway to assess its efficacy in patients with intermediate- to high-risk or relapsed/refractory MF [118]. A key safety concern is capillary leak syndrome, which is highlighted in its black box warning [118].

The second mitochondrial activator of caspase (SMAC) is a protein that promotes apoptosis by interacting with cellular inhibitor of apoptosis proteins (cIAPs) [119]. SMAC mimetics are small molecules engineered to replicate this function, effectively neutralizing cIAPs and restoring apoptotic signaling, particularly in cells resistant to tumor necrosis factor (TNF)-induced cell death. By promoting caspase activation, they enhance the apoptotic response. Overexpression of SMAC has been associated with increased sensitivity to apoptosis, making this an attractive therapeutic target in resistant disease [120]. The oral SMAC mimetic LCL161 has demonstrated antineoplastic activity in MPN models both in vitro and in vivo, including significant spleen size reduction in a JAK2V617F-driven murine model. In a phase II clinical trial involving patients with primary MF, post-PV MF, and post-ET MF, LCL161 achieved an objective response in 38% of patients. The median time to response was 1.4 months, and the treatment duration reached up to 55.2 months in some individuals [121].

Figure 3 provides an illustration of current and future perspective therapies in MPNs.

Lysyl oxidases are copper-dependent amine oxidases that catalyze the oxidative deamination of lysine residues, promoting the cross-linking of collagen and elastin by generating reactive aldehydes. This enzymatic activity plays a role in tissue remodeling and fibrosis [122]. A previous study highlighted the involvement of LOX in the pathogenesis of MF in murine models [123]. Additionally, pan-LOX inhibition using small molecules was shown to reduce splenomegaly and bone marrow fibrosis in MF mouse models, supporting the potential therapeutic value of targeting LOX activity in MF [124].

PXS-5505, a pan-lysyl oxidase inhibitor, has demonstrated strong and sustained inhibition of lysyl oxidases, with favorable pharmacokinetic/pharmacodynamic (PK/PD) profiles and excellent tolerability in healthy volunteers at doses exceeding 100 mg daily for up to 14 days. In MF patients, plasma LOXL2 levels appear elevated compared to healthy individuals, suggesting its potential utility as a disease biomarker—especially given its lower baseline presence in circulation compared to LOX, which is widely expressed in other tissues. Although initial analysis showed no clear association between LOX or LOXL2 levels and specific MPN subtypes in a small cohort, further evaluation is ongoing [125].

Table 3 summarizes perspective and emerging therapeutic strategies in MPNs beyond conventional JAK Inhibitors and Interferon-α2b.

## 9. Conclusions

The evolving understanding of the immunopathology of MPNs reveals that chronic inflammation, immune evasion, and stromal dysfunction are central drivers of disease progression and therapeutic resistance. The differential expression and function of key cytokines—such as GRO-α, IL-23, and TGF-β—highlight the complexity and heterogeneity within MPN subtypes. Dysregulated crosstalk between monocytes, platelets, megakaryocytes, and the bone marrow microenvironment shapes the pro-inflammatory and pro-fibrotic state that characterizes advanced disease.

Emerging therapies targeting immune checkpoints, chemokine signaling (CXCR1/2), TGF-β pathways, and erythropoietic regulators offer promising avenues beyond JAK inhibition. Among these, agents such as ruxolitinib-combination immunotherapies, TGF-β traps (e.g., AVID200), and luspatercept have already entered early-phase clinical trials, reflecting the closest translational readiness to clinical practice. In contrast, strategies like PD-L1 vaccines, CXCL1/CXCR1-2 blockade, LOX inhibitors, and SMAC mimetics remain at the preclinical or early exploratory stage, requiring further safety and efficacy evaluation before clinical adoption. The therapeutic potential of these mechanism-based interventions underscores a shift toward more precise, translationally guided approaches in MPN management.

## Figures and Tables

**Figure 1 jcm-14-06328-f001:**
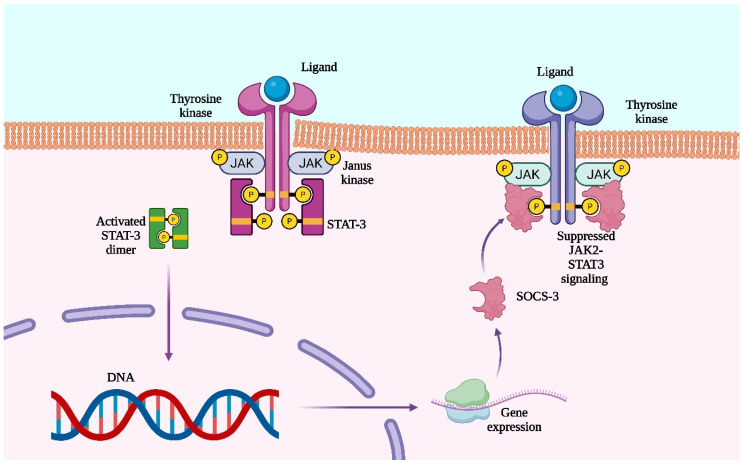
JAK2–STAT3 signaling pathway and its negative feedback regulation via SOCS-3. Upon ligand binding to a tyrosine kinase receptor, JAK (Janus kinase) phosphorylates STAT-3 (Signal Transducer and Activator of Transcription 3), leading to dimerization and nuclear translocation of activated STAT-3, where it drives gene expression. One of the transcribed genes is SOCS-3 (Suppressor of Cytokine Signaling 3), which inhibits JAK2–STAT3. “P” indicates phosphorylation. Generated with Biorender.com.

**Figure 2 jcm-14-06328-f002:**
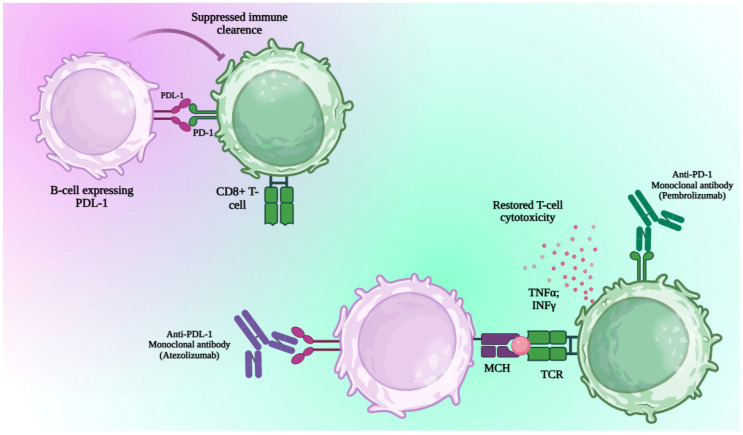
Immune Checkpoint Blockade Restores CD8^+^ T Cell Function via PD-1/PD-L1 Inhibition; interaction between PD-1 (Programmed Cell Death Protein 1) on CD8^+^ T cells (cytotoxic T lymphocytes) and PD-L1 (Programmed Death Ligand 1) on B cells or malignant cells suppresses immune clearance by inducing T cell exhaustion. This immune suppression can be reversed through checkpoint blockade using anti–PD-L1 monoclonal antibodies (e.g., Atezolizumab) or anti–PD-1 monoclonal antibodies (e.g., Pembrolizumab), which restore T cell cytotoxic function. Reactivated CD8^+^ T cells recognize antigen via the TCR (T Cell Receptor) interacting with MHC (Major Histocompatibility Complex) and produce TNF-α (Tumor Necrosis Factor Alpha) and IFN-γ (Interferon Gamma), enhancing anti-tumor immunity. Generated with Biorender.com.

**Figure 3 jcm-14-06328-f003:**
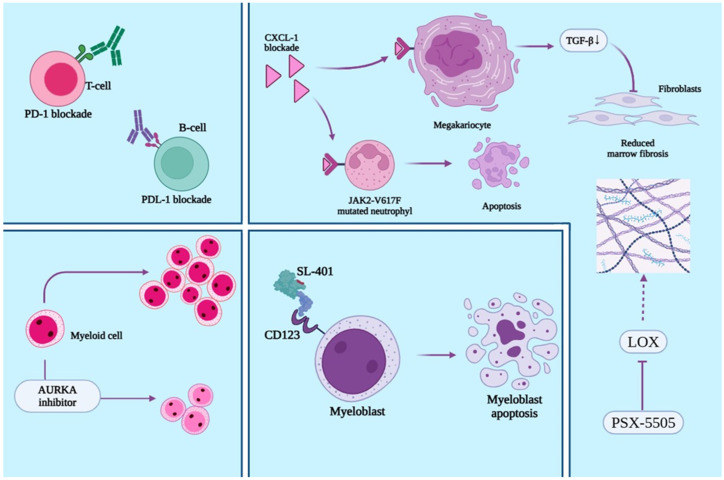
Novel therapeutic targets and future treatment strategies in myeloproliferative neoplasms (MPNs). Checkpoint inhibition through PD-1 (programmed death-1) and PD-L1 (programmed death-ligand 1) blockade aims to restore T and B cell-mediated immunity. CXCL1 blockade (e.g., reparixin) in JAK2V617F-mutated neutrophils restores normal apoptosis and in megakaryocytes reduces TGF-β (transforming growth factor beta) production, leading to decreased marrow fibrosis via inhibition of fibroblast activation. Aurora kinase A (AURKA) inhibitors promote apoptosis in abnormal myeloid cells (especially in megakariocytes). SL-401 (tagraxofusp) targets CD123 on myeloblasts, inducing apoptosis. LOX (lysyl oxidase) inhibition with PXS-5505 disrupts collagen cross-linking and fibrosis development. These strategies reflect a multipronged approach targeting immune evasion, fibrosis, inflammation, and malignant hematopoiesis in MPNs. Continuous line arrow- means stimulatory effect, continuos arrow with blunt head means inhibitory effect, and dash arrow means a reduction/inhibition in activity. Generated with Biorender.com.

**Table 1 jcm-14-06328-t001:** Overview of immune cells alterations in myeloproliferative neoplasms and their modulation under treatment.

Immune Cells	In MPNs	Under Treatment	Study Type	Reference
CD4^+^ T cells	↑ Activation, ↑ PD-1 expression, effector shift	↓ PD-1^+^ cells, restoration toward resting phenotype	Clinical, prospective	[34]
CD8^+^ T cells	↑ PD-1^+^ exhausted phenotype, ↓ proliferation and cytotoxic function	↓ PD-1^+^ cells, improved viability and function	Clinical, prospective	[34]
Regulatory T cells (Tregs)	↓ Frequency, dysfunctional, contributes to chronic inflammation	↓ Further decreased with JAK inhibitors, shift toward Th17	Clinical, prospective	[37,38,40,41]
Th17 cells	↑ Pro-inflammatory, potential tumor-promoting or anti-tumor roles	↑ Dominant profile post-JAK inhibition	Clinical, prospective	[38,39,41]
PD-L1^+^ B cells (Bregs)	↑ Abundant, immunosuppressive, resistant to anti-CD20	↓ With ruxolitinib or ruxolitinib + IFNα2b	Clinical, retrospective	[32,35,36]
PD-1^+^ B cells	↑ Immunoregulatory, suppress T-cell proliferation via PD-L1	↓ After therapy	Clinical, retrospective	[33,35]
NK cells	↓ Number and cytotoxic function	↓ NK cell numbers due to impaired maturation;	Clinical, cross-sectional	[49,52]
Dendritic cells (DCs)	↓ Frequency, impaired monocyte-to-DC differentiation	Further suppression under ruxolitinib, with ↓ secretion of IL-12 and IL-15	Clinical, cross-sectional	[42,43]
Innate Lymphoid Cells (ILCs)	JAK2: ↑ ILC1;	Mutation-dependent modulation; effect not fully reversed	Clinical, cross-sectional	[42,43]

Bregs: Regulatory B cells; DCs: Dendritic cells; IFNα2b: Interferon alpha-2b; ILC1: Group 1 innate lymphoid cells; ILCs: Innate lymphoid cells; JAK: Janus kinase; MPNs: Myeloproliferative neoplasms; NK cells: Natural killer cells; PD-1: Programmed cell death protein 1; PD-L1: Programmed death-ligand 1; T cells: T lymphocytes; Th17: T helper 17 cells; Tregs: Regulatory T cells; ↑: increase in number/frequency; ↓: decrease in number/frequency.

**Table 2 jcm-14-06328-t002:** Cytokine and chemokine patterns in myeloproliferative neoplasms.

Category	Cytokine/Factor	MPN Subtype	Matrix Used	Key Findings	References
Pro-inflammatory cytokines	IL-2, IL-2R, IL-6	PMF	Plasma	Reflect T-cell activation and chronic inflammation	[73]
	IL-12, IL-17, TNF-α, IFN-α	PMF	Plasma	Promote Th1/Th17 cell differentiation and immune dysregulation	[74,75]
	IL-23	PV	Plasma	Supports Th17 maintenance; selectively elevated in PV	[81,82]
Anti-inflammatory cytokines	IL-1RA, IL-4, IL-10	PMF	Plasma	Upregulated; IL-10 inhibits myelopoiesis (CFU-GM inhibition)	[73,74,76]
Fibrosis-associated cytokines	TGF-β, FGF, TPO	PMF	Plasma	Drive marrow fibrosis and remodeling	[73,79]
	IL-13	PMF	Plasma from humans, BM aspirate and plasma from mice	Enhances TGF-β expression, collagen biosynthesis; expands mutant megakaryocytes	[80]
Chemokines	MIP-1β, RANTES, MCP-1	PMF	Plasma	Promote monocyte recruitment and stromal activation	[73,74,77]
	MCP-1 -2518 A/G (SNP)	PMF (sMF)	Peripheral blood	Linked to severe disease; germline variation affects chemokine expression and fibrosis	[78]
	GRO-α (CXCL1)	ET ↑, PMF ↓	Blood serum	Elevated in ET; paradoxically low in PMF	[88,89]
Angiogenic factors	VEGF	PV, PMF	Peripheral blood	Increased in PV and PMF; promotes eNOS, HIF-1α, especially in PV	[83,84,85]
	IL-6 + VEGF	PV	Peripheral blood	VEGF–IL-6 positive feedback loop contributes to vascular remodeling	[85]
Growth factors	HGF, IL-11	PV	Blood serum and BM aspirate	Promote STAT3 activation; mutation-independent mechanism supporting JAK2V617F cell growth	[86,87]
Hematopoietic support	SCF (Stem Cell Factor)	ET (↓)	BM aspirate	Low SCF impairs CD34^+^ HSPC expansion, BM microenvironment, and Treg differentiation	[90]

BM: Bone marrow; CXCL1 (GRO-α): Growth-regulated oncogene alpha; ET: Essential thrombocythemia; FGF: Fibroblast growth factor; HGF: Hepatocyte growth factor; IFN-α: Interferon alpha; IL: Interleukin; IL-1RA: Interleukin-1 receptor antagonist; IL-2R: Interleukin-2 receptor; IL-6: Interleukin-6; IL-10: Interleukin-10; IL-11: Interleukin-11; IL-12: Interleukin-12; IL-13: Interleukin-13; IL-17: Interleukin-17; IL-23: Interleukin-23; MCP-1: Monocyte chemoattractant protein-1; MPN: Myeloproliferative neoplasm; PMF: Primary myelofibrosis; PV: Polycythemia vera; RANTES: Regulated on Activation, Normal T cell Expressed and Secreted; SCF: Stem cell factor; sMF: Secondary myelofibrosis; SNP: Single nucleotide polymorphism; STAT3: Signal transducer and activator of transcription 3; TGF-β: Transforming growth factor beta; TNF-α: Tumor necrosis factor alpha; TPO: Thrombopoietin; VEGF: Vascular endothelial growth factor; ↑: arrow up; ↓: arrow down.

**Table 3 jcm-14-06328-t003:** Overview of Novel Therapeutic Agents in Myeloproliferative Neoplasms.

Agent	Mechanism of Action	Clinical Status	Findings	Reference
Checkpoint inhibitors				
Pembrolizumab	Anti–PD-1 checkpoint blockade	Phase II, single-arm—negative; study stopped after stage 1	No objective responses; immune profiling suggests enhanced T-cell activity without clinical benefit; well tolerated	[102]
Nivolumab		Phase II—negative/terminated early	No objective responses; 5/8 stable disease (median 3.3 mo); median OS 6.1 mo; study stopped early	[103]
PD-L1 peptide vaccine	Induces anti-regulatory T-cell response	Preclinical	71% displayed spontaneous PD-L1–specific T-cell responses; stronger in non-advanced vs. advanced MPN; responses mainly CD4+, no safety assessment	[104]
CXCR1/2 inhibition				
Reparixin	CXCR1/2 inhibitor; reduces TGF-β in megakaryocytes	Preclinical	Reduced BM and splenic fibrosis in GATA1^low^ mice	[105]
Aurora kinase inhibitors				
Alisertib	Inhibits Aurora kinase A	Phase I; safety and preliminary efficacy demonstrated	Reduced splenomegaly and symptom burden; normalized atypical megakaryocytes; decreased BM fibrosis in sequential biopsies; partial reduction in allele burden	[112]
Others				
TGF-β trap	Target fibrosis-inducing cytokines	Phase Ib	Modest clinical benefit (spleen, symptoms, progenitor restoration)	[113,114]
Luspatercept	Enhances erythropoiesis	Phase II	Improved hemoglobin and reduced transfusion burden; symptom improvement; stable spleen size	[115]
Sotatercept	Enhances erythropoiesis	Phase II	Increased Hb in non-TD; achieved transfusion independence in RBC-TD; effective as monotherapy and in combination with ruxolitinib	[116]
Tagraxofusp	Targets CD123 (IL-3Rα) on malignant cells	Phase I/II	27 MF patients treated; 53% spleen reduction in evaluable patients, 45% symptomatic response; better responses in patients with monocytosis;	[118]
SMAC mimetics (LCL161)	SMAC mimetic	Preclinical	JAK2V617F-mutant cells hypersensitive; reduces splenomegaly and potentially fibrosis in mice; effect dependent on JAK2 kinase and NFĸB; exogenous TNFα or JAK inhibition alters efficacy	[121]
PXS-5505 (pan-LOX inhibitor)	Pan-lysyl oxidase inhibitor	Phase I	Well tolerated; achieved strong plasma LOX inhibition; PK/PD supports daily dosing;	[125]

BM (Bone marrow), CXCR1/2 (C-X-C chemokine receptor types 1 and 2), Hb (Hemoglobin), IL-3Rα (Interleukin 3 receptor alpha), LOX (Lysyl oxidase), MF (Myelofibrosis), MPN (Myeloproliferative neoplasms), PD-1 (Programmed cell death protein 1), PD-L1 (Programmed death-ligand 1), PK/PD (Pharmacokinetics/pharmacodynamics), SMAC (Second mitochondrial activator of caspases), TD (transfusion dependent), TGF-β (Transforming growth factor beta).

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
