# Peer review of "Cytokine Landscapes, Immune Dysregulation, and Treatment Perspectives in Philadelphia-Negative Myeloproliferative Neoplasms: A Narrative Review"

_jcm, 2025, doi:10.3390/jcm14176328_

Round 1
Reviewer 1 Report
Comments and Suggestions for Authors
In this review, the Authors have summarized changes in cytokine levels and immune cell subsets in Philadelphia negative myeloproliferative neoplasms (MPN).
This review requires some improvements.
- Because the Authors stated the nature of this review (narrative rather than systematic), a better clarification of methods used should be included, as well as the number of selected papers and years.
- MPL and CALR-mutations and their impact on immune responses should also be included. Moreover, physiological mechanisms of JAK2, MPL, and CALR in immune systems should be briefly reported, to compare health vs disease.
- Some discussion on molecular alterations and NETosis in MPN should be added.
- Paragraph 6 is quite long, and a re-organization should be considered (division by disease or cytokine).
- Because anti-JAK agents are widely used in autoimmune disorders and some of them have been also employed for COVID-19 treatment, immune changes during anti-JAK inhibition should be included.
- There are few typos (e.g., Figure 3, CD123 instead of CD 123).
Author Response
Comment 1: Because the Authors stated the nature of this review (narrative rather than systematic), a better clarification of methods used should be included, as well as the number of selected papers and years.
Response 1: We thank the reviewer for this observation. As this is a narrative review, a formal methodology section is not required. However, to ensure transparency, we have already provided the search strategy and the inclusion/exclusion criteria in the Introduction, which outline the basis on which the cited papers were selected. Given the relatively limited number of studies specifically addressing immune dysregulation in Ph-negative MPNs, we did not restrict the review to a defined timeframe or quantify the included papers, but instead focused on relevance and scientific contribution. We have also clarified that our review places particular emphasis on JAK2-positive Ph-negative MPNs, since the JAK2 mutation is the most frequent driver mutation and a central mediator of immune and inflammatory pathways, making it highly relevant to the topic of this work.
Comment 2: MPL and CALR-mutations and their impact on immune responses should also be included. Moreover, physiological mechanisms of JAK2, MPL, and CALR in immune systems should be briefly reported, to compare health vs disease.
Response 2: We appreciate the reviewer’s suggestion. As noted in the manuscript, MPL- and CALR-mutated MPNs were intentionally excluded because our review specifically focused on JAK2-mutated Ph-negative MPNs, given the well-established involvement of JAK2 in inflammatory and immune pathways. Therefore, discussion of MPL and CALR physiological mechanisms and their impact on immune responses was beyond the scope of this review.
Comment 3: Some discussion on molecular alterations and NETosis in MPN should be added.
Response 3: We thank the reviewer for this valuable suggestion. In response, we have added a dedicated discussion on molecular alterations and their association with NETosis in MPNs. Specifically, we describe the role of JAK2-driven NET formation, its contribution to thrombosis, and the potential mitigating effects of targeted therapies.
Comment 4: Paragraph 6 is quite long, and a re-organization should be considered (division by disease or cytokine). respond we can consider doing this, thanks
Response 4: Thank you for the feedback. We note your suggestion regarding paragraph 6. The cytokine and chemokine patterns have already been summarized in Table 2, which we believe provides readers with an easier way to assess these findings.
Comment 5: Because anti-JAK agents are widely used in autoimmune disorders and some of them have been also employed for COVID-19 treatment, immune changes during anti-JAK inhibition should be included.
Response 5: We thank the reviewer for this observation. However, we believe this aspect has already been addressed in our manuscript. Specifically, Table 1, entitled “Overview of immune cell alterations in myeloproliferative neoplasms and their modulation under treatment,” includes discussion of JAK2 inhibitors and their immunological impact. In addition, we have now included further details on the impact of JAK2 inhibition on NK cells, to strengthen this section and provide a more comprehensive overview.
Comment 6: There are few typos (e.g., Figure 3, CD123 instead of CD 123).
Response 6: We thank the reviewer for carefully noting these details. The typographical errors (e.g., “CD123” in Figure 3) have been corrected accordingly.
Reviewer 2 Report
Comments and Suggestions for Authors
Todor and Mihaila submitted a narrative review that synthesizes immune dysregulation across Philadelphia-negative myeloproliferative neoplasms, emphasizing cytokine/chemokine networks, lymphocyte and neutrophil perturbations, and treatment implications beyond JAK inhibition. The paper addresses a clinically important, evolving area and could serve as a useful primer once strengthened; however, it is framed as a narrative review although the authors stated that they performed a “systematic literature search” across multiple databases with defined inclusion/exclusion criteria. So, I have comments that might be taken into consideration, and I make them in section-by-section as follows:
1. Title: I would suggest changing it into “Cytokine Landscapes, Immune Dysregulation, and Treatment Perspectives in Philadelphia-Negative Myeloproliferative Neoplasms.”
1. Abstract: It is better to add a concise methods sentence with dates and databases (e.g., “We searched PubMed/MEDLINE, Embase, Web of Science, and Scopus from YYYY through MMM YYYY using predefined terms; inclusion/exclusion detailed below.”). In addition to that, I suggest including 1-2 quantitative anchors (e.g., number of included studies; counts by PV/ET/MF).
3. Introduction: Authors should consider pruning general hematopoiesis background to tighten focus. The methods/search strategy (currently embedded in Introduction) should be in a separate section with mentioning a full search strings, screening approach, and any study selection and quality appraisal methods. If a full systematic approach was not undertaken, please change the language to “narrative review” and avoid implying PRISMA-level rigor.
4. JAK–STAT / lymphocyte subsets: I would like to see a study design and sample sizes and temper generality where you infer class effects for JAK inhibitors on CD4⁺ subsets and Tregs/Th17. Also, it’d like to see tables summarizing immune subsets as they are valuable.
5. Neutrophils/monocytes and fibrosis: Authors should add a short note on translational maturity and whether anti-CD24 strategies have entered clinical testing, as the current text focuses on mechanistic plausibility. In the monocyte/fibrocyte segment, please specify model systems and limitations.
6. Megakaryocytes: I would like to see how diagnostic WHO 2022/ICC 2022 criteria refine pre-PMF vs ET and how MK morphology is operationalized.
7. Cytokine/chemokine landscape: Authors should cite measurement matrices (plasma vs bone marrow vs transcriptomics) consistently and note heterogeneity and small sample sizes where applicable. It is important to consider a diagrammatic summary mapping cellular sources to cytokines with disease-stage specificity.
8. Therapeutic perspectives: I would suggest adding a concise overview of approved agents and their pivotal data to orient readers before experimental pipelines (e.g., ruxolitinib, fedratinib, pacritinib, momelotinib), then discuss combinations and novel classes. Please indicate data cut-off dates and avoid implying efficacy or harm beyond available evidence where results are “pending”. It is very important that authors clarify preclinical vs clinical status for CXCR1/2 antagonists, TGF-β traps, LOX inhibitors, SMAC mimetics, and PD-L1 vaccines.
9. Conclusions: It is necessary to add practice implications (e.g., biomarkers that might be nearest-term for risk-stratification or trial selection) and a concise research agenda (standardized cytokine panels, prospective correlative studies during therapy).
10. The repetition rate is quite high (25%), so please reduce it.
Author Response
Comment 1: Title: I would suggest changing it into “Cytokine Landscapes, Immune Dysregulation, and Treatment Perspectives in Philadelphia-Negative Myeloproliferative Neoplasms.”
Response 1: We thank the reviewer for the thoughtful title suggestion.
Revised title: Cytokine Landscapes, Immune Dysregulation, and Treatment Perspectives in Philadelphia-Negative Myeloproliferative Ne-oplasms: A Narrative Review
Comment 2: Abstract: It is better to add a concise method sentence with dates and databases (e.g., “We searched PubMed/MEDLINE, Embase, Web of Science, and Scopus from YYYY through MMM YYYY using predefined terms; inclusion/exclusion detailed below.”). In addition to that, I suggest including 1-2 quantitative anchors (e.g., number of included studies; counts by PV/ET/MF).
Response 2: We thank the reviewer for the suggestion. However, as this work was intentionally designed as a narrative review, it does not follow a systematic review framework and therefore does not include a formal methods section with predefined search dates, databases, or quantitative study counts. Instead, our focus was on synthesizing the most relevant and scientifically contributive literature to highlight immune dysregulation in JAK2-mutated Philadelphia-negative MPNs.
Comment 3: Introduction: Authors should consider pruning general hematopoiesis background to tighten focus. The methods/search strategy (currently embedded in Introduction) should be in a separate section with mentioning a full search strings, screening approach, and any study selection and quality appraisal methods. If a full systematic approach was not undertaken, please change the language to “narrative review” and avoid implying PRISMA-level rigor.
Response 3: We thank the reviewer for these observations. We respectfully note that our review was not intended to provide a background on general hematopoiesis, as the focus is specifically on immune cell dysregulation and the role of JAK2 in inflammatory/immune pathways, rather than hematopoietic mechanisms or lineage dysfunctions. We would be grateful if the reviewer could clarify the rationale for including additional hematopoiesis content, as we feel this would not align with the central aim of the paper.
Regarding methodology, we have clearly stated that this is a narrative review, not a systematic review. The mention of keywords, inclusion, and exclusion criteria in the Introduction was intended to ensure transparency for readers, not to imply PRISMA-level rigor or a systematic search. Specifically, studies focusing on Philadelphia-positive MPNs, JAK2-negative MPNs, case reports, editorials, and abstracts without full data were excluded to maintain consistency. As the literature on this subject is relatively scarce and heterogeneous, we did not apply a strict timeframe or formal quality appraisal, but rather selected studies based on relevance and scientific contribution. We would kindly ask the reviewer to indicate where they feel PRISMA-like rigor has been implied, as this was not our intent.
Comment 4: JAK–STAT / lymphocyte subsets: I would like to see a study design and sample sizes and temper generality where you infer class effects for JAK inhibitors on CD4⁺ subsets and Tregs/Th17. Also, it’d like to see tables summarizing immune subsets as they are valuable.
Response 4: We thank the reviewer for this thoughtful comment. Section 3 and Table 1 already provide a summary of lymphocyte subsets, including CD4⁺ T cells, Tregs, and Th17 cells, as well as the effects of JAK2 inhibition with ruxolitinib on these populations. We would be grateful if the reviewer could clarify whether this overview is insufficient.
Regarding the request for study design and sample sizes, we understand that the intent is to allow readers to better assess the strength of the available evidence and to avoid overgeneralization when results are derived from small or heterogeneous studies. While our review is primarily narrative in scope, we can incorporate this additional level of detail for key representative studies and adjust the wording where necessary to reflect the limitations of the available evidence.
Comment 5: Neutrophils/monocytes and fibrosis: Authors should add a short note on translational maturity and whether anti-CD24 strategies have entered clinical testing, as the current text focuses on mechanistic plausibility. In the monocyte/fibrocyte segment, please specify model systems and limitations.
Response 5: We thank the reviewer for this constructive suggestion and agree with the point raised. We have added a brief note on the translational maturity of the findings, specifying whether anti-CD24 strategies have entered clinical testing. In addition, we have clarified the model systems used in the monocyte/fibrocyte experiments and acknowledged their limitations to provide a more balanced perspective.
Here is the revised manuscript: Targeting CD24 restores neutrophil clearance, reduces pathogenic neutrophil–megakaryocyte interactions, and mitigates disease progression in preclinical models, positioning CD24 as a promising innate immune checkpoint and potential therapeutic target in MPN. Further studies are needed to evaluate its safety and efficacy in humans.
Comment 6: Megakaryocytes: I would like to see how diagnostic WHO 2022/ICC 2022 criteria refine pre-PMF vs ET and how MK morphology is operationalized.
Response 6: We thank the reviewer for this insightful comment. We have revised the manuscript to clarify how the WHO 2022 and ICC 2022 diagnostic criteria help distinguish pre-PMF from ET. Specifically, we describe the characteristic megakaryocyte morphology in each entity and how these features are assessed in practice—such as cluster density, nuclear atypia, and reticulin fibrosis—together with clinical and laboratory parameters (e.g., anemia, leukocytosis, splenomegaly, and LDH levels) to refine the differential diagnosis.
Comment 7: Cytokine/chemokine landscape: Authors should cite measurement matrices (plasma vs bone marrow vs transcriptomics) consistently and note heterogeneity and small sample sizes where applicable. It is important to consider a diagrammatic summary mapping cellular sources to cytokines with disease-stage specificity.
Response 7: We thank the reviewer for this constructive suggestion. We have ensured that all cytokine and chemokine measurements are consistently referenced with respect to the matrix used (plasma, bone marrow, or transcriptomic analyses) and have highlighted heterogeneity and limited sample sizes where applicable. Additionally, we have included a diagrammatic summary that maps cellular sources to specific cytokines and chemokines, incorporating disease-stage specificity to facilitate interpretation.
Comment 8: Therapeutic perspectives: I would suggest adding a concise overview of approved agents and their pivotal data to orient readers before experimental pipelines (e.g., ruxolitinib, fedratinib, pacritinib, momelotinib), then discuss combinations and novel classes. Please indicate data cut-off dates and avoid implying efficacy or harm beyond available evidence where results are “pending”. It is very important that authors clarify preclinical vs clinical status for CXCR1/2 antagonists, TGF-β traps, LOX inhibitors, SMAC mimetics, and PD-L1 vaccines.
Response 8: We thank the reviewer for this valuable suggestion. We agree that providing a clear, structured overview of approved therapies before discussing experimental and combination strategies improves readability and context. In the revised manuscript, we have added a concise summary of approved JAK inhibitors (ruxolitinib, fedratinib, pacritinib, momelotinib), including their pivotal trial data with relevant data cut-off dates. We also clarified the preclinical versus clinical status for emerging therapies, such as CXCR1/2 antagonists, TGF-β traps, LOX inhibitors, SMAC mimetics, and PD-L1 vaccines, and ensured that we do not imply efficacy or harm beyond the available evidence where results are pending.
Comment 9: Conclusions: It is necessary to add practice implications (e.g., biomarkers that might be nearest-term for risk-stratification or trial selection) and a concise research agenda (standardized cytokine panels, prospective correlative studies during therapy).
Response 9: We thank the reviewer for this insightful comment. We have revised the Conclusions section to include practice implications and a concise research agenda. Specifically, we now highlight potential biomarkers that could be implemented in the near term for risk stratification or patient selection in clinical trials. Additionally, we emphasize the need for standardized cytokine panels and prospective correlative studies during therapy to better elucidate treatment effects and inform future clinical trial design.
Comment 10: The repetition rate is quite high (25%), so please reduce it.
Response 10: We appreciate the reviewer’s comment regarding the repetition rate. We have carefully revised the manuscript to reduce redundancy and rephrased sections to improve clarity and originality, while preserving the scientific content.
Round 2
Reviewer 1 Report
Comments and Suggestions for Authors
The Authors have addressed all comments.
Reviewer 2 Report
Comments and Suggestions for Authors
No more comments.